# Assessing Public Willingness to Wear Face Masks during the COVID-19 Pandemic: Fresh Insights from the Theory of Planned Behavior

**DOI:** 10.3390/ijerph18094577

**Published:** 2021-04-26

**Authors:** Muhammad Irfan, Nadeem Akhtar, Munir Ahmad, Farrukh Shahzad, Rajvikram Madurai Elavarasan, Haitao Wu, Chuxiao Yang

**Affiliations:** 1School of Management and Economics, Beijing Institute of Technology, Beijing 100081, China; irfan@ncepu.edu.cn or haitao.kungfuer@gmail.com (H.W.); yangchuxiao1991@hotmail.com (C.Y.); 2Center for Energy and Environmental Policy Research, Beijing Institute of Technology, Beijing 100081, China; 3School of Urban Culture, South China Normal University, Nanhai Campus, Foshan 528225, China; 4Pakistan Center, North Minzu University, Yinchuan 750001, China; 5School of Economics, Zhejiang University, Hangzhou 310058, China; munirahmad@zju.edu.cn; 6School of Economics and Management, Guangdong University of Petrochemical Technology, Maoming 525000, China; farrukh.hailian@gmail.com; 7Clean and Resilient Energy Systems (CARES) Laboratory, Texas A&M University, Galveston, TX 77553, USA; rajvikram787@gmail.com

**Keywords:** willingness to wear, face masks, COVID-19, theory of planned behavior, risk perceptions

## Abstract

Face masks are considered an effective intervention in controlling the spread of airborne viruses, as evidenced by the 2009′s H1N1 swine flu and 2003′s severe acute respiratory syndrome (SARS) outbreaks. However, research aiming to examine public willingness to wear (WTW) face masks in Pakistan are scarce. The current research aims to overcome this research void and contributes by expanding the theoretical mechanism of theory of planned behavior (TPB) to include three novel dimensions (risk perceptions of the pandemic, perceived benefits of face masks, and unavailability of face masks) to comprehensively analyze the factors that motivate people to, or inhibit people from, wearing face masks. The study is based on an inclusive questionnaire survey of a sample of 738 respondents in the provincial capitals of Pakistan, namely, Lahore, Peshawar, Karachi, Gilgit, and Quetta. Structural equation modeling (SEM) is used to analyze the proposed hypotheses. The results show that attitude, social norms, risk perceptions of the pandemic, and perceived benefits of face masks are the major influencing factors that positively affect public WTW face masks, whereas the cost of face masks and unavailability of face masks tend to have opposite effects. The results emphasize the need to enhance risk perceptions by publicizing the deadly effects of COVID-19 on the environment and society, ensure the availability of face masks at an affordable price, and make integrated and coherent efforts to highlight the benefits that face masks offer.

## 1. Introduction

The outbreak of novel coronavirus (COVID-19) has become a significant public health issue worldwide [1]. The pandemic has severely influenced 216 countries in total and had an unprecedented impact on peoples’ daily routines [2]. As of 8 April 2021, the number of positive COVID-19 cases reached 133.8 M, with 2.90 M global deaths [3]. Government efforts to combat the virus have been made through extensive diagnostic tests and recommendations on social distancing with an aim to prioritize human health [4]. The rigorous social distancing measures were first implemented in China [5]. On 26 February 2020, the Pakistan Ministry of Health reported the first confirmed COVID-19 case in Karachi. Another case was confirmed in Islamabad on the same day by the Ministry of Health [6]. The number of COVID-19 positive cases increased to twenty within the next fifteen days, with Sindh Province having the highest number of cases, followed by Gilgit Baltistan Province. All these cases were found to have a recent travel history from Syria, London, and Iran [7]. At present, the number of cases is increasing at an alarming rate and the situation is worsening. According to official statistics, the total number of confirmed COVID-19 cases in Pakistan has reached 705,517, with 15,124 deaths [8]. To prevent the spread of the virus, social distancing practices and online social presence were reported to play an integral role [9].

The Pakistani government has taken several initiatives in the form of isolation centers, authorized hospitals, testing facilities, case tracing, and risk communication to limit the spread of COVID-19 in the country. For instance, the Pakistan Ministry of National Health Services, Regulation & Coordination issued the “National Action Plan for Preparedness & Response to COVID-19” to limit positive cases and strengthen the state by providing an appropriate and effective response to possible events caused by the epidemic (SARS-CoV-2) [10]. In addition, strict restrictions, i.e., quarantine and social distancing policies, were also imposed. These restrictions have seriously affected all economic activities of the country. [11].

Antiviral medication is believed to be the best shield against the novel SARS-CoV-2 in terms of reducing morbidity and mortality. However, the vaccine development process has taken time, and the supply may be inadequate. Alternatively, there are ways to minimize the spread of COVID-19 until the availability of a vaccine. For instance, face masks have been strictly utilized to fight airborne viruses, including the 2003 SARS coronavirus (SARS-CoV) [12] and the 2009 H1N1 swine flu virus [13]. Moreover, face masks are cost-effective compared with other nonpharmaceutical interventions [14].

Some researchers have identified the impact of meteorological factors (temperature, humidity) and air pollutants (NO_2_, PM_2.5_, PM_10_, SO_2_, ozone) on COVID-19 spread [15,16]. In addition, a few have investigated the indirect environmental effects of the COVID-19 pandemic [5,15]. Others have explored and debated post-pandemic behaviors [17,18]. However, studies assessing public willingness to wear (WTW) face masks in response to the COVID-19 outbreak are scarce and have been conducted mostly in western parts of the world. Taking this debate into account, the present work intends to respond to this research gap by conducting a comprehensive study in Pakistan. This paper is the first of its kind to examine public WTW face masks by considering the following two research questions: (i) What are the possible influencing factors that may encourage or discourage Pakistani people from wearing face masks in response to the COVID-19 outbreak? (ii) How do these influencing factors shape public WTW face masks? Moreover, we have expanded the behavioral framework of theory of planned behavior (TPB) [19] by integrating three novel dimensions to deepen academic analyses of the COVID-19 pandemic. By identifying the influencing factors and how they shape public willingness, this research can help government institutions and policymakers develop robust policies for the prevention of pandemics.

The remainder of the paper is structured as follows: Section 2 explains the methods and hypothesis formulation. Section 3 illustrates the research design. Study results are reported in Section 4. Discussion of the research results are presented in Section 5. Lastly, Section 6 concludes the study and offers policy recommendations.

## 2. Methods

### 2.1. Theoretical Framework

Consumers’ willingness to buy a certain product is a complex process that involves a variety of factors [20]. In order to understand the dynamic nature of consumers’ buying process, a variety of theoretical frameworks are employed by various scholars. For instance, some pioneer theories include self-efficacy theory (SET), social cognitive theory (SCT), the theory of reasoned action (TRA), and TPB [21,22]. Albert Bandura, a psychologist, proposed SET [23]. It is defined as individuals’ beliefs in their ability to exert control over their own functioning and over life events. Self-efficacy can act as a springboard for inspiration, well-being, and personal achievement. The four primary sources of influence on people’s beliefs about their effectiveness are (i) mastery experiences, (ii) vicarious experiences, (iii) social persuasion, and (iv) emotional states. Albert Bandura founded Social Learning Theory (SLT) in 1960. Later, it evolved into the SCT in 1986, which asserts that learning occurs in a social context characterized by complex and reciprocal interactions between the person, the environment, and behavior [24]. SCT is distinctive in that it emphasizes on social influence and its importance on external and internal social reinforcement. SCT takes into account the particular way in which individuals learn and sustain actions, as well as the social context in which individuals behave. The theory recognizes a person’s previous interactions, which determine whether or not behavioral activity will occur. These prior experiences form reinforcements, perceptions, and expectancies, which all influence whether or not an individual will engage in a particular behavior and the reasons for engaging in that behavior [25].

Fishbein and Ajezen developed TRA in 1975 [26]. However, TRA focused more on voluntary actions based on personal attitudes and subjective norms. Although the action is not always voluntary, it is still under supervision. To tackle this situation, Ajzen proposed TPB [19]. TPB stipulates that behavioral intentions govern people’s behavior [27]. A behavior is executed once people weigh the consequences of their actions, leading to a needed result. Different from contextual studies, we have focused on public WTW face masks in this research. Researchers have widely adopted TPB in the healthcare domain to describe and forecast public behavior regarding their WTW face masks [28]. They believe that TPB more-successfully scrutinizes public behavior than other theories. Therefore, TPB is utilized to develop the research framework of this study, following the previous literature [29]. The description of the original TPB model is displayed in Figure 1.

Three factors constitute behavioral intention. These are: (i) attitudes towards the behavior, (ii) subjective norms, and (iii) perceived behavioral control. A person’s general feeling of favorableness or unfavorableness for a particular behavior is termed as their attitude towards the behavior [31]. People’s attitudes are shaped by their striking convictions and the outcomes associated with a specific behavior [32], while the total sum of beliefs about a product by prominent individuals and groups make up subjective norms, and they think that an individual should follow this behavior and comply with them [33]. Perceived behavioral control is defined as individuals’ opinions of how easy or challenging it is to perform a behavior of interest based on one’s perceived enablers or impediments to that behavior [34] (see Figure 1). TPB has stimulated a significant volume of empirical health behavior research. Researchers have assumed that numerous elements influence the acceptance of a particular product or service in social, economic, and political terms [35,36]. Moreover, people are concerned about the perceived risk of the pandemic, the perceived benefits of face masks, and the unavailability of face masks. Therefore, we have advanced the structural framework of TPB by incorporating three novel dimensions. With the inclusion of these dimensions, this framework assists in examining public WTW face masks comprehensively. Figure 2 depicts the research framework of this paper.

### 2.2. Formulation of Hypotheses

#### 2.2.1. Attitude

Attitude (ATD) is a vital element of TPB that is described as a person’s positive or negative assessment of a specific behavior [19,37]. In behavioral medicine, it is considered buyers’ favorable or unfavorable response to community health problems. Walter et al. [38] reported that research on ATD during pandemic situations not only directs mitigation strategies but also provides an opportunity for future pandemic preparedness planning. In addition, ATD is a critical factor that influences peoples’ decisions on whether to accept personal protective equipment (PPE), i.e., face masks, respirators, gloves, protective clothing, goggles, and hand sanitizers. Previous studies have reported that there is a positive relationship between attitude and WTW face masks. Zhang and Mu [39] found that people have a positive attitude that exposure to heavy air pollution can be reduced by wearing face masks. Johnson and Hariharan [40] examined the impact of attitude on face mask wearing behavior during the H1N1 swine flu pandemic. Their findings revealed that respondents exhibited a high level of WTW face masks. In light of these findings, the first hypothesis was proposed as follows:

**Hypothesis** **1.***Attitude positively influences public willingness to wear face masks*.

#### 2.2.2. Social Norms

There is often a perceived social obligation to perform a specific behavior [41]. Social norms (SNR) are considered the influence of family, friends, neighbors, and peers on WTW face masks. Previous scholars have shown that face mask wearing is positively influenced by SNR. Santana et al. [42] found that social norms motivate COVID-19 preventive behaviors such as wearing face masks. Syed et al. [43] revealed that households’ willingness of wearing face masks considerably increased during the 2003 SARS outbreak and was positively linked with SNR. In another study, Taylor et al. [44] reported that Australian respondents who were extremely worried about family and friends during the H1N1 swine flu pandemic exhibited more WTW face masks. Burgess and Horii [45] conducted a survey in Japan and revealed that the majority of Japanese people believed that people should be respectful of the health concerns of other individuals by wearing face masks. Consequently, mask wearing is considered a social obligation in Japan. Overall, SNR have a significant influence on individuals’ intention to wear face masks. We therefore presumed that the similar effect would be observed in the current study and formulated as follows:

**Hypothesis** **2.***Social norms positively influence public willingness to wear face masks*.

#### 2.2.3. Cost of Face Masks

Cost information is an important attribute in relation to the economic losses related to the buying process [46]. The outcomes of many studies have confirmed the negative association between the cost of face masks (CST) and public WTW face masks. Weiss and Palmer [47] examined the association between the cost of face masks and low literacy levels and found that cost is the main barrier to buying face masks. Kesselheim [48] analyzed the relationship between high face mask costs and life-cycle management. The findings revealed that high costs increase the strain of patients, leading to adverse health effects by decreasing adherence to necessary medications. Although the costs of PPE and healthcare items have declined during the last decade, they still cost more than the affordability of most people living in developing countries [49,50]. These research outcomes allowed us to devise the third hypothesis as follows:

**Hypothesis** **3.***Cost of face masks negatively influences public willingness to wear face masks*.

#### 2.2.4. Risk Perceptions of the Pandemic

Risk perceptions of the pandemic (RPP) positively contribute to shaping public WTW face masks. Public WTW face masks increases when individuals perceive their susceptibility to the pandemic and its severity. If the risk of infection is perceived as high, a quicker public response would be formed in terms of adopting protective behaviors [28]. The outcomes of former studies have revealed that risk perceptions play a critical role in shaping individuals’ decisions to accept PPE. Several researchers have indicated that the social acceptance of face masks is positively influenced by risk perceptions. For instance, MacIntyre and Chughtai [51] analyzed the factors affecting WTW face masks among Chinese adults and reported that risk perceptions positively affected public willingness. Similarly, Barati et al. [52] examined public behavior concerning the acceptance of face masks to prevent respiratory infection. Their results revealed that the risk perceptions of being infected with acute diseases persuade individuals to wear face masks. Considering these outcomes, we devise the fourth hypothesis as follows:

**Hypothesis** **4.***Risk perceptions of the pandemic positively influence public willingness to wear face masks*.

#### 2.2.5. Perceived Benefits of Face Masks

People’s understanding and awareness of the benefits that face masks offer in controlling and preventing the transmission of infectious viral diseases is termed perceived benefits of face masks (PBFM) [53]. People compare the performance of face masks with conventional preventive methods and decide according to the effectiveness of face masks as a social health measure [54]. They perceive that wearing face masks minimizes the spread of the virus from infected to healthy individuals in public gatherings. In addition, face masks remind people to practice social distancing measures [55]. Hansstein and Echegaray [56] assessed the motivations behind wearing face masks among young Chinese adults and found that as the air quality has worsened, awareness among the Chinese population of climate issues and health consequences has rapidly increased. Consequently, individuals have formed positive beliefs regarding the benefits of wearing face masks. Moreover, accessibility and convenience of use have further strengthened their beliefs in favor of face masks. Thus, we devised the fifth hypothesis, considering the above arguments, as follows:

**Hypothesis** **5.***The perceived benefits of face masks positively influence public willingness to wear face masks*.

#### 2.2.6. Unavailability of Face Masks

If an individual is not capable of performing a specific behavior, the corresponding intentions will not occur. The unavailability of face masks (UFM) is related to people’s difficulty in obtaining them [57]. The effort associated with the use of face masks is one attribute that could affect public willingness. People should be given access to key resources for the acceptance and utilization of face masks [58]. The outcomes of former studies have revealed that UFM plays a nonsignificant role in individuals’ choices to wear face masks. Several researchers have indicated that WTW face masks is negatively influenced by UFM. For instance, Tang and Wong [59] analyzed the factors affecting WTW face masks among Chinese adults and reported that UFM is a major barrier that negatively affects their willingness. Similarly, Maclntyre et al. [53] examined public behavior concerning the acceptance of face masks in preventing respiratory infections. Their results revealed that low WTW face masks is associated with their unavailability, which renders them ineffective for controlling seasonal respiratory diseases. Finally, the primary reason for UFM in developing countries is the high cost, making the encouragement of public willingness a difficult task [60]. These arguments led us to devise the sixth hypothesis as follows:

**Hypothesis** **6.***The unavailability of face masks negatively influences public willingness to wear face masks*.

## 3. Research Design

### 3.1. Survey Site, Sample Size, and Selection of Respondents

An inclusive questionnaire survey was administered in the provincial capitals of Pakistan, including Lahore, Peshawar, Karachi, Gilgit, and Quetta, during 2020. Lahore is the provincial capital of Punjab Province; Peshawar is the provincial capital of Khyber Pakhtunkhwa Province; and Karachi, Gilgit, and Quetta are the provincial capitals of Sind, Gilgit Baltistan, and Baluchistan Provinces, respectively (see Figure 3). The fundamental rationale in selecting these provincial capitals as survey sites is that the respondents to be surveyed belonged to heterogeneous communities in these diverse provinces of Pakistan. Another reason for the selection of the survey sites was that these provincial capitals have distinctive characteristics and have a greater number of COVID-19 patients than other areas of the country.

Before conducting the survey, the authors visited the provincial capitals of Pakistan to identify the distinguishing features of participants living in these cities. Then, the respondents were approached in person (contacted personally) for the actual questionnaire survey [61,62,63]. The following criteria were considered for the selection of respondents. (i) The respondents should be permanent residents of these cities. (ii) The age of the respondents should be not less than 18 years. Responses were generated using the convenience sampling method [64,65,66,67], meaning that the sampling process was not purely randomized due to the ongoing epidemic. Generally, this sampling method, due to convenience and feasibility, is useful for researchers in certain special situations, such as epidemics or experimental behavioral research. Therefore, the empirical findings based on the selected sample may not be perfectly generalizable. However, in the current case, the respondents’ demographic features show that the questionnaires were conducted among respondents of heterogeneous backgrounds. Thus, the findings generated based on such a sample provide a fair representation of the population with heterogeneous backgrounds in terms of education, age, income, and occupation. Moreover, the questionnaire survey was conducted in all of Pakistan’s provincial capital cities and involved respondents from populations with heterogeneous cultures and diverse behaviors. Therefore, the generated sample was rich enough to satisfactorily represent a population with heterogeneous features.

The questionnaire process was divided into two phases. During the first phase, the questionnaires were administered to 900 respondents, and they were allowed a time period of one month to complete their responses. A detailed description was given to the respondents about every element of the questionnaire to obtain accurate and meaningful results. During the second phase, the questionnaires were returned by the respondents after one month. A total of 738 valid responses were collected, for a response rate of 82% [68,69]. The following three criteria were applied to consider a response valid: (i) All aspects of the questionnaire were thoroughly completed. (ii) The questionnaire had no missing or incomplete information. (iii) Finally, the questionnaire did not have multiple responses. The description of the survey is provided in Table 1. Comfrey and Lee [70] recommended the following scale to determine the adequacy of sample size: (very poor—50), (poor—100), (fair—300), (very good—500), (excellent—1000 or more). According to this scale, the size of our study sample (738 respondents) falls under the “very good” category, ensuring that the sample is representative for this research.

### 3.2. Selection of Variables

The work of Hung [54] was accessed to determine the scale items for measuring “attitude”. The scale items measuring “social norms” were taken from [56,71]. The scale items associated with “risk perceptions of the pandemic” were acquired from the research of [28], while those related to “unavailability of face masks” were compiled from the analysis of [57]. The scale items for measuring “perceived benefits of face masks” and “cost of face masks” were taken from the research of [59,72], respectively. Finally, the scale items associated with “public WTW face masks” were taken from the work of [54]. A five-point Likert scale was employed to assess each item, with 1 specifying “strongly disagree” and 5 specifying “strongly agree” (see Table A1 of Appendix A).

### 3.3. Statistical Analyses

SPSS (V. 26) (IBM, New York, NY, USA) and Amos (V. 26) software (IBM, New York, NY, USA) package were utilized for performing exploratory factor analysis (EFA), confirmatory factor analysis (CFA), structural equation modelling (SEM), and testing the proposed hypotheses. SEM is a frequently employed technique due to its flexibility and generality. It comprises of several steps, including specification, estimation, evaluation, and modification of the model. The technique is robust for investigating the relationship among multiple variables and have numerous benefits over common multivariate approaches: (i) a reliable assessment of measurement errors, (ii) valuation of latent variables by observed variables, and (iii) model checking for the evaluation and implementation of a framework based on data consistency [73]. In addition, the majority of multivariate methods implicitly neglect measurement error. However, SEM computes variables by taking into consideration the measurement errors [74]. Due to these advantages, SEM produces reliable and valid results [75]. Consequently, we employed SEM, as it is the most successful technique to scrutinize the association among all the selected factors.

## 4. Results

### 4.1. Demographic Features of the Respondents

Table 2 reports the demographic characteristics of the respondents. Most of the respondents (325, 44%) belonged to the middle-age cohort, followed by the young cohort (232, 31.4%). The old-age cohort (181, 24.5%) was the third-largest group. There were more males (387, 52.4%) than females in our sample. Two hundred forty-seven respondents (33.5%) belonged to the middle-income class, having a per-month income between USD 201 and 300, followed by the lower-middle-income class (218, 29.5%) with a per month income between USD 101 and 200. Moreover, we classified the sample in various education levels: 270 (36.6%) had a college degree, whereas 192 (26%) had a high school education. In our survey, 322 (43.6%) of the respondents had a technical occupation.

### 4.2. Descriptive Statistics and Discriminant Validity Findings

The descriptive statistics were scrutinized by means and standard deviations. Pearson’s correlation analysis was conducted to test the interrelationships among the variables. The analysis generated significant correlations among the variables. The discriminant validity was investigated using the root square of the average variance extracted (AVE). The results supported discriminant validity because the root square of AVE was higher than its correlation with other variables [76]. The results are disclosed in Table 3.

### 4.3. Testing the Fit of the Model

To assess the consistency of all variable elements, a composite reliability (CR) test was performed. In addition, convergent validity was investigated using AVE and item loadings [77]. The outcomes confirmed that the values of AVE for each factor exceeded 0.50, emphasizing that the latent variables maintained more than 50% variance. Sample reliability was examined using reliability analysis. The results showed that for all variables, the values of CR and Cronbach’s α exceeded the least accepted value of 0.70 (see Table 4), as suggested by [78]. All these findings confirmed the validity and reliability of data.

EFA was carried out to obtain the causal design structure. The Kaiser–Meyer–Olkin (KMO) and Bartlett’s sphericity tests (BTS) were performed before EFA to measure the fit of the data. The KMO value was 0.917, indicating that we could proceed with factor analysis [79]. Similarly, BTS generated a significant value of 9406.783, which fulfilled the condition for EFA (see Table 5). Next, CFA was executed to scrutinize the appropriateness of the data for the proposed research framework. The content validity of the measurement model was confirmed, as all items were significantly loaded on their respective constructs (see Figure 4).

### 4.4. Testing of Hypotheses and Structural Equation

After determining that our measures were valid and reliable, the authors tested the proposed model and the hypothesized relationships. The *R*^2^ value was computed as an essential step to determine the variation in the outcome variable explained by the explanatory variables. The *R*^2^ value was 0.74, which was higher than the minimum recommended value of 0.35 [80], implying significant interpretation. We performed the covariance-based curve estimation and SEM algorithm to inspect the linkages in the model. The analysis provided a strong *F*-value, indicating linearity among all the relationships. Then, a collinearity diagnostic analysis was conducted to examine the issue of multicollinearity. The recommended variance inflation factor (VIF) value must not be greater than 10 [81]. The findings indicated that the model did not have a multicollinearity issue because the VIF values are within the suggested value and supported by the findings of [82].

Figure 5 displays the path diagram of SEM. Three significance levels were considered, such as 1%, 5%, and 10%. Significance at 1% level (*p* ≤ 0.001) is indicated by (***), significance at 5% level (*p* ≤ 0.01) is indicated by (**), while significance at 10% level (*p* ≤ 0.05) is indicated by (*). The path coefficients of the variables “attitude”, “social norms”, “risk perceptions of the pandemic”, and “perceived benefits of face masks”, *H1* (b = 0.09, *p* < 0.01), *H2* (b = 0.11, *p* < 0.01), *H4* (b = 0.65, *p* < 0.01), and *H5* (b = 0.09, *p* < 0.05), respectively, specify that ATT, SNO, RPP, and PBFM have significant and positive impacts on public WTW face masks. Thus, hypotheses 1, 2, 4, and 5 were accepted. On the other hand, willingness decreases with increases in face mask costs and the unavailability of face masks, as the variables “cost of face masks” *H3* (b = −0.00, *p* < 0.001) and “unavailability of face masks” *H6* (b = −0.10, *p* < 0.01) negatively affect public WTW face masks. Accordingly, hypotheses 3 and 6 were also accepted. Table 6 illustrates the structural paths and the validity of hypotheses. Different fitness tests were also applied to confirm whether the data was adequately fit for the proposed model. The findings (reported in Table 7) reveal that all fit index values are in line with the recommended criteria [83].

### 4.5. Endogeneity Testing

This test is used primarily to determine the robustness of the study findings [84]. Endogeneity bias may distort the estimate of maximum probability, which is a significant challenge to the acceptability of the findings. The Heckman test was performed in Stata software to solve this issue and examine endogeneity. The results (presented in Table 8) showed significance similar to that of the previous model, suggesting that endogeneity bias does not exist in our findings.

## 5. Discussion

### 5.1. Attitude and WTW Face Masks

The findings supported the hypothesis that ATD positively affects public WTW face masks, which indicates that people who are thoroughly familiar with the COVID-19 pandemic have a higher tendency to wear face masks. The former research of [39] highlighted that attitude plays a vital role because people exhibit an optimistic attitude that wearing face masks could reduce the probability of being infected by viral respiratory diseases. Similarly, the study of [40] showed that attitude has a favorable impact on public WTW face masks. The findings of these studies comply with our results. Owing to the current global pandemic situation, majority of the citizens have recognized that the usage of face masks can tackle the spread of COVID-19 and help to solve the health dilemma. The regrettable development is that the novel pandemic is growing in Pakistan, which will have significant effects on future public WTW face masks.

### 5.2. Social Norms and WTW Face Masks

The results further highlighted that SNR have a positive effect on public WTW face masks. The results are in line with the previous studies of [53,85], as they reported that public WTW face masks is positively affected by social norms. One major reason is possibly that Pakistani society is well integrated, and inputs from neighbors, family, and friends have a strong and lasting impact on people’s minds [34]. Thus, SNR play a leading role in decision making. The understanding of wearing face masks may affect public behavior in such a manner that a positive experience encourages people to wear face masks. Rahim et al. [86] conducted a survey in seven universities of Pakistan and found that 60% of the participants highlighted the need to use face masks, gloves and other PPE to protect from respiratory infections.

### 5.3. Cost of Face Masks and WTW Face Masks

The likelihood of public WTW face masks decreases with the additional price associated with buying face masks. The results supported our hypothesis, as cost negatively affects public willingness. Previous research findings supported our results, as Kesselheim [48] found that cost had a negative impact on public intentions to use face masks, and cost was a major obstacle to accepting new advances in the health sector. Chughtai and Khan [57] found that several factors contribute to the selection and use of face masks, such as cost, presence of adverse events, and pre-existing medical illness. Similarly, Weiss et al. [47] noted that public willingness was influenced by cost and that high cost prevented individuals from buying face masks. One likely reason might be that face masks are cheaper in most advanced countries, such as China, the USA, Germany, and France, than in Pakistan. A middle-class family in Pakistan cannot afford extra costs and does not dare to purchase face masks. In this regard, a rise in healthcare expenditures will not only reduce the costs of protective equipment such as face masks but will also improve living standards [87,88].

### 5.4. Risk Perceptions of the Pandemic and WTW Face Masks

Our findings reveal that RPP positively influences public WTW face masks. Previous findings confirmed the role of risk perceptions in shaping public behavior during pandemics [52,53] and are parallel with our research results. Munir et al. [28] conducted a study in China to scrutinized the perception-based influence factors of individuals’ intention to adopt COVID-19 epidemic prevention. The authors found that risk perception has a positive impact on people’s intentions to practice epidemic prevention. It implies that increasing people’s awareness of the infection’s severity, susceptibility, and fatality will increase their intention to adopt epidemic prevention measures. Hamamura and Park [89] compared face mask-wearing behavior among American, Chinese, and Japanese respondents. The findings revealed that Chinese and Japanese people tend to wear face masks more often while going out than American people. The possible factors that motivate people to wear face masks include perceived reduced chances of being infected with SARS-CoV-2 and controlling the spread of airborne diseases. The stronger the perceptions of the lethal aspects of the pandemic are, the easier it would be to influence public willingness to wear face masks. These risk perceptions develop more confidence in wearing face masks and can help as an important dynamic in the future.

### 5.5. Perceived Benefits of Face Masks and WTW Face Masks

The hypothesis results indicate that PBFM has a significant effect on public WTW face masks. These findings are consistent with former studies that found that the purchasing decisions of individuals are established on the optimistic belief in the effects of a specific product that they intend to buy [13,57]. MacIntyre and Chughtai [51] conducted a study to assess the efficacy of face masks against coronaviruses for the community, healthcare workers and sick patients. They researchers revealed that community mask usage is beneficial and very important during the COVID-19 pandemic in universal community face mask use as well as in health care settings. Face masks should be worn continuously during a shift, according to trials in healthcare workers. This could help prevent COVID-19 infections and deaths among health workers. People show WTW face masks if they recognize the perceived advantages of their usage in terms of fewer chances of getting infected [90]. One main reason might be that as the awareness of Pakistani people about COVID-19 is increasing, they are developing positive beliefs about how face masks will help them surmount the pandemic.

### 5.6. Unavailability of Face Masks and WTW Face Masks

The results revealed that public WTW face masks is negatively influenced by UFM, and this finding is supported by [91]. The possible factors that could discourage people from wearing face masks include the struggle to obtain face masks in particular areas where individuals reside. In addition, they think that drugstores charge a high price for face masks due to the low supply in the market, which is a leading factor of public unwillingness to wear face masks. Chughtai and Khan [57] reported that during SARS and the H1N1 pandemic of 2009, the shortage of face masks along with other PPE was the primary reason for negative behavior towards the acceptance of face masks. In addition, technological shifts during pandemic outbreaks can play an influential role in restructuring society [92]. The pandemic also induced a behavioral shift in power sector operations around the world [93]. Moreover, as the acceptance and usage of face masks are in preliminary stages in the country, people are reluctant to accept them.

### 5.7. Demographic Factors and WTW Face Masks

In addition to the proposed influencing factors, some demographic factors also affect public WTW face masks. For instance, Bish and Michie [94] studied the impact of demographic determinants on public WTW face masks and found that public willingness was significantly influenced by age and gender. The results revealed that older people and females exhibit more protective behavior than other groups of society. Condon and Sinha [95] obtained similar results, as females showed more willingness to use face masks than their male counterparts during the 2009 H1N1 swine flu pandemic.

### 5.8. Summary and Limitations of Research

Among the positive contributors to WTW face masks, risk perception was the most substantial contribution. The more the risk is perceived, the more people will be willing to use face masks. Thus, a lack of risk perception might lead to contradictory behavior. Therefore, steady efforts to make people aware of the pandemic’s fatality and lethality will continue to improve risk perception, positively impacting WTW face masks. In contrast, the public attitude and perceived benefits of face masks remained the weakest contributors to promoting WTW face masks. These findings imply that people respond to the benefits of face masks with lower intensity than to the threats of not wearing them amid pandemic outbreaks such as COVID-19. Among the negative contributors, the cost of face masks made an almost negligible but significant contribution. This finding indicated the importance of cost in shaping consumers’ purchase decisions regarding face masks. In contrast, the unavailability of face masks proved to be a negative and relatively stronger contributor to public willingness to wear them compared to the cost of face masks. It depicted the actual scenario of the country. Amid the pandemic outbreak, a shortage of face masks was observed across the country. Hence, the unavailability of face masks poses a practical obstacle. To overcome this hurdle, the supply of face masks should be enhanced to increase people’s willingness to purchase and wear them.

There are some limitations of the current study that should be taken into account in future research. In the face of the pandemic outbreak, this research opted for a convenience sampling technique, potentially causing sampling bias. This situation limits the perfect generalizability of the empirical findings to the whole population. The Kolmogorov–Smirnov test between the population and the sample has been employed by previous studies, such as [96], to deal with this issue. They categorized construction workers based on only one demographic feature: age. However, in the present case, the application of this test is not feasible because the population data for heterogeneous demographic features are unavailable to compare with the sampled respondents’ features. Therefore, future studies should conduct a randomized sampling technique to make the results more generalizable to the whole population. Another limitation of the current study is that a sample size of 738 is not enough for several cities for the generalizability of the findings. However, it is not possible to expand the sample size at this stage, when the survey has been completed and analysis had already been done. Subsequent studies can tackle this limitation by expanding the sample size in the same and other geographical locations. The present research has expanded the TBP in a theoretical setting, which is not a concise way to model the desired factors. Therefore, future studies should incorporate the factors by developing a mathematical or statistical model to provide a brief and precise picture of modeled factors.

## 6. Conclusions

This study assesses public WTW face masks in response to the COVID-19 pandemic by analyzing the factors that affect the willingness of Pakistani people to wear face masks. As a step further, we expanded the structural framework of TPB by incorporating three novel dimensions (risk perceptions of the pandemic, perceived benefits of face masks, and unavailability of face masks) to comprehensively analyze all the possible behavioral factors that may inspire people to wear face masks or prevent them from wearing them. An inclusive survey was conducted in the provincial capitals of Pakistan, and data analysis was performed by employing SEM. The results indicate that attitudes, social norms, risk perceptions of the pandemic, and perceived benefits of face masks have positive and significant effects on public WTW face masks. In contrast, cost of face masks and unavailability of face masks were found to have a prohibiting effect.

The research outcomes indicate that influencing factors, i.e., attitude, subjective norms, risk perceptions of the pandemic, and the perceived benefits of face masks, positively affect public WTW face masks. Therefore, policy makers should pay close attention to these factors in their efforts to successfully shape public willingness to wear face masks. In addition, it should be emphasized that it is necessary to reform the national education curriculum so that children start prioritizing environmental values from childhood so that these practices will later lead to favorable effects on society. The government should repeatedly inform the masses to stay at home and wear face masks when going out to avoid the spread of SARS-CoV-2. The utilization of social, print, and electronic media to highlight the lethal aspects of COVID-19 would be helpful in this regard.

The findings also reveal the strongest and weakest contributors to public WTW face masks based on the degree of their specific contributions. Among the positive contributors, risk perception made the strongest contribution. At the same time, attitude and benefits of face masks made the weakest contributions. In contrast, the unavailability of face masks revealed the strongest negative contribution to public WTW face masks. However, the weakest negative contribution was the cost of face masks.

Face masks are somewhat costly in the country due to a low supply, and drugstores charge high prices, making affordability very difficult for a low-income family. The government should ensure the availability of face masks at an affordable price, provide subsidies and financial incentives to poor people, and carefully monitor prices on a regular basis. Drugstores that charge more than the set price should be fined, and their licenses should be canceled. In addition, the government should formulate a robust policy to make it compulsory for all people to wear face masks during the pandemic to actively eradicate the spread of COVID-19.

The research results revealed that face mask costs and unavailability are critical barriers to public willingness to wear them; therefore, robust policy development is needed to overcome these impediments. To this end, on the one hand, the federal and provincial governments should subsidize the import of face masks. On the other hand, the availability of face masks should be ensured by systematically monitoring local medical stores. An alternative option to reduce consumer face mask prices could be the subsidization of local manufacturers. Rapid domestic production of face masks will curtail their cost for end-users. Moreover, the enhanced provision of face masks is expected to increase willingness to wear them. The emergence of the novel COVID-19 has compelled people to follow certain laws in order to prevent its transmission among the public. One of these laws is the obligation to wear a facemask at public places. As such, the demand for face masks has escalated recently. Pakistan is a developing country with a high population density. To meet the demand for its large population, the country is facing numerous challenges such as limited manufacturing capacity of face masks, lack of certified suppliers and sellers, low quality of locally manufactured face masks and reliance on foreign countries for the import of face masks. R&D is a vital value-adding segment of the health industry’s value chain. From the perspective of future pandemics, the government of Pakistan should invest resources in R&D to innovate face mask production materials that would incur lower costs, making the availability of low-cost face masks feasible, and satisfy the current needs of ever-increasing population. In this regard, the government should work in collaboration with private institutions and manufacturing companies and devise a solution, which achieves the following three important requirements: (i) The facemasks should be inexpensive. (ii) The facemasks should be sturdy and safe. (iii) The facemasks should be washable, sterilizable, and reusable. One classic example is the manufacturing of a three-dimensional reusable facemask, which is being used in several countries. A three-dimensional reusable facemask has the following characteristics: (i) It can be conveniently made on a low-cost, non-heated bed at a low temperature. (ii) It uses a small quantity of filament content. (iii) It can be washed and disinfected, making it reusable. (iv) It uses small quantities of disposable non-woven cloth, which should be discarded after each use [97]. Another such example is the development of innovative masks (such as degradable masks, reusable masks, and antiviral masks). However, this is impossible to achieve without government support in the form of allocating special funds and subsidies for R&D activities in the long-run [98]. Additionally, risk perception strongly drove peoples’ willingness to wear face masks; therefore, the pandemic’s lethality and fatality should be communicated at all levels of society to enable people to more accurately perceive the pandemic’s risk. In this way, the enhanced credibility of the pandemic threat will promote willingness to wear face masks.

## Figures and Tables

**Figure 1 ijerph-18-04577-f001:**
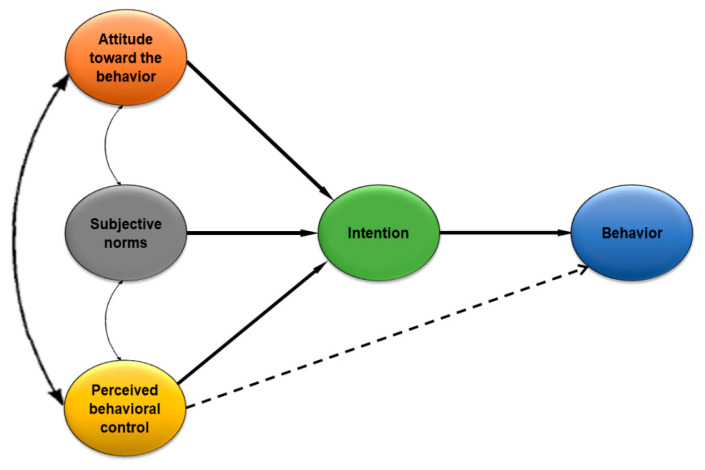
Description of the original TPB model [30].

**Figure 2 ijerph-18-04577-f002:**
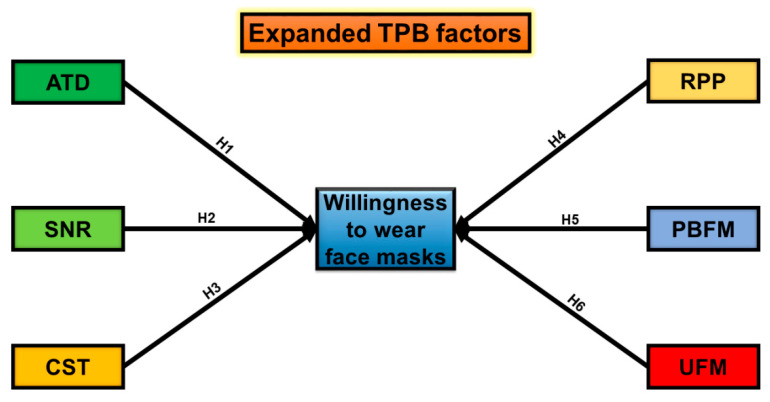
Research framework presenting the influencing factors of public WTW face masks. Notes: ATD: Attitude, SNR: Social norms, CST: Cost of face masks, RPP: Risk perceptions of the pandemic, PBFM: Perceived benefits of face masks. UFM: Unavailability of face masks.

**Figure 3 ijerph-18-04577-f003:**
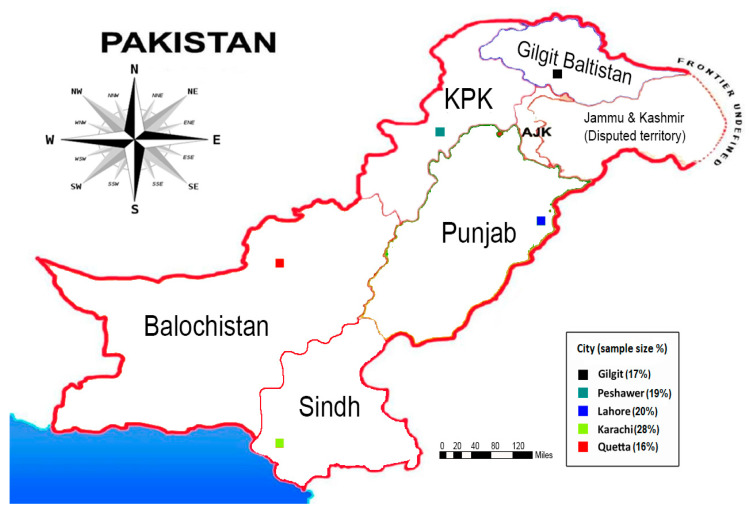
Survey sites.

**Figure 4 ijerph-18-04577-f004:**
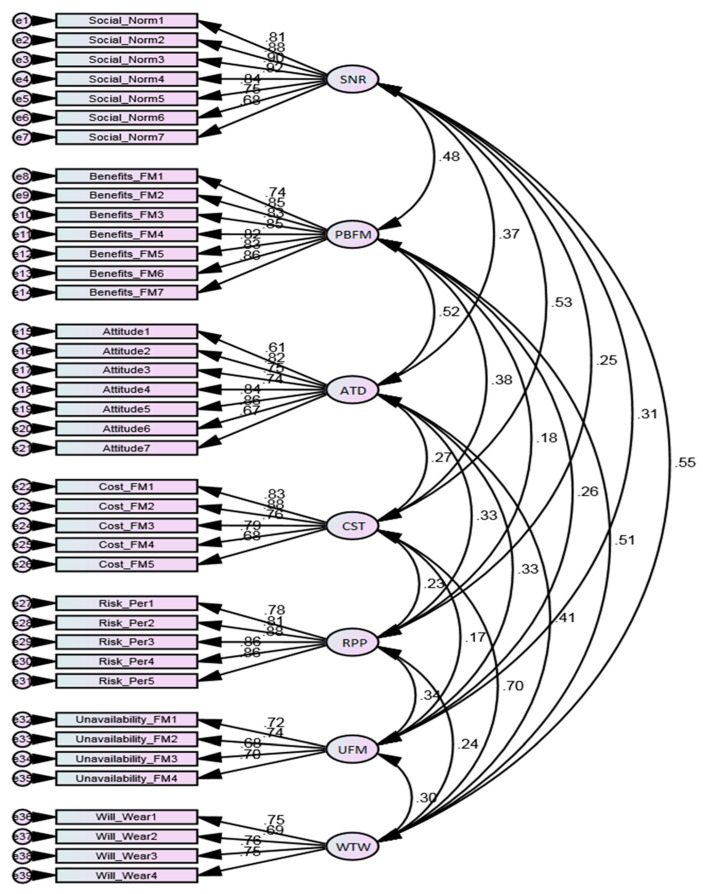
Measurement model. Notes: All items are loaded on their respective constructs, confirming the content validity of the measurement model. The model also supports discriminant validity, as the outer loading for all constructs are less than 0.80. Convergent validity is validated as well, as the inner loadings for all constructs are greater than 0.70.

**Figure 5 ijerph-18-04577-f005:**
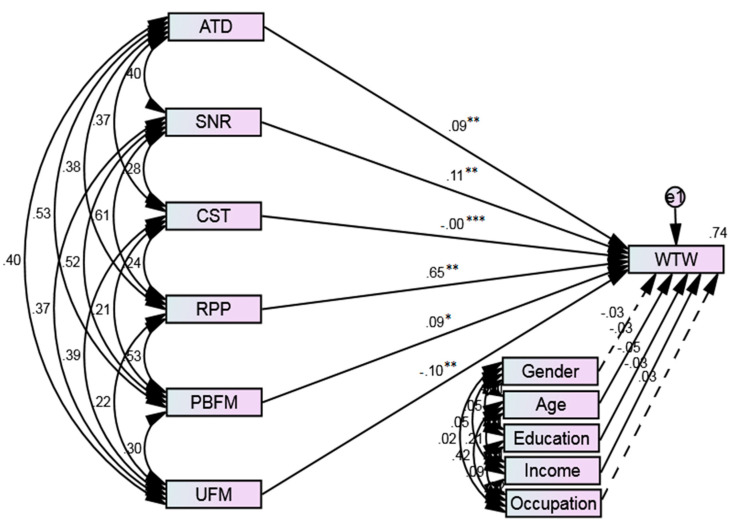
Path diagram of SEM. Solid lines represent significant paths, while dashed lines represent insignificant paths. *** *p* ≤ 0.001 (1%), ** *p* ≤ 0.01 (5%), * *p* ≤ 0.05 (10%).

**Table 1 ijerph-18-04577-t001:** Description of survey.

Parameters	Value
Time frame	August, September, and October (2020)
Location of the survey	Lahore, Peshawar, Karachi, Gilgit, and Quetta
Size of the sample	900
Valid responses	738
Response rate	82%

**Table 2 ijerph-18-04577-t002:** Demography of the respondents.

Features	Options	Frequencies	(%)
Age	18–35	232	31.4
	36–55	325	44
	Above 55	181	24.5
Gender			
	Male	387	52.4
	Female	351	47.6
Income (USD)			
	<100	39	5.3
	101–200	218	29.5
	201–300	247	33.5
	301–400	167	22.6
	>400	67	9.1
Education			
	Uneducated	32	4.3
	Primary	106	14.4
	High school	192	26
	College pass	270	36.6
	Post-graduation	138	18.7
Occupation			
	Government job	32	4.3
	Technical worker	322	43.6
	Entrepreneur	206	27.9
	Other	178	24.1

**Table 3 ijerph-18-04577-t003:** Correlation and test of discriminant validity.

Factors	UFM	SNR	PBFM	ATD	CST	RPP	WTW
UFM	(0.711)						
SNR	0.326	(0.824)					
PBFM	0.267	0.491	(0.822)				
ATD	0.354	0.375	0.523	(0.753)			
CST	0.171	0.545	0.417	0.305	(0.777)		
RPP	0.341	0.256	0.181	0.329	0.224	(0.836)	
WTW	0.296	0.571	0.507	0.417	0.724	0.242	(0.738)

Notes: Diagonal values represent the root square of AVEs.

**Table 4 ijerph-18-04577-t004:** Factor loadings and results of reliability, composite reliability, and convergent validity.

Factors	Items	Outer Loadings	AVE	CR	Cronbach-α
Attitude			0.567	0.901	0.903
	ATD1	0.562			
	ATD2	0.834			
	ATD3	0.722			
	ATD4	0.659			
	ATD5	0.898			
	ATD6	0.907			
	ATD7	0.615			
Social norms			0.679	0.936	0.938
	SNR1	0.774			
	SNR2	0.800			
	SNR3	0.940			
	SNR4	0.969			
	SNR5	0.830			
	SNR6	0.705			
	SNR7	0.651			
Cost of face masks			0.604	0.884	0.891
	CST1	0.884			
	CST2	0.975			
	CST3	0.688			
	CST4	0.672			
	CST5	0.513			
Risk perceptions of the pandemic			0.699	0.921	0.918
	RPP1	0.729			
	RPP 2	0.798			
	RPP 3	0.902			
	RPP 4	0.864			
	RPP 5	0.869			
Perceived benefits of face masks			0.675	0.936	0.937
	PBFM1	0.641			
	PBFM2	0.837			
	PBFM3	0.803			
	PBFM4	0.860			
	PBFM5	0.851			
	PBFM6	0.818			
	PBFM7	0.899			
Unavailability of face masks			0.506	0.804	0.803
	UFM1	0.729			
	UFM 2	0.747			
	UFM 3	0.681			
	UFM 4	0.674			
Willingness to wear face masks			0.545	0.827	0.824
	WTW1	0.658			
	WTW2	0.691			
	WTW3	0.662			
	WTW4	0.608			

Notes: Cumulative variance explained: 63.92%, Rotation method: Promax with Kaiser normalization, Extraction method: Maximum likelihood.

**Table 5 ijerph-18-04577-t005:** Kaiser–Meyer–Olkin (KMO) and Bartlett’s test.

KMO and Bartlett’s Test		
Kaiser-Meyer-Olkin Measure of Sampling Adequacy		0.817
Bartlett’s Test of Sphericity	Approx. Chi-Square	9406.783
	df	78
	Sig.	0.000

Notes: Sig: Significance, df: Degree of freedom.

**Table 6 ijerph-18-04577-t006:** Results of hypotheses.

**Hypotheses**	**Structural Paths**	**b Value**	**Result**	**VIF**	***R*^2^**
*H1*	ATD ➝ WTW	0.09 **	Accepted	1.631	0.74
*H2*	SNR ➝ WTW	0.11 **	Accepted	1.811	
*H3*	CST ➝ WTW	−0.00 ***	Accepted	1.281	
*H4*	RPP ➝ WTW	0.65 **	Accepted	1.375	
*H5*	PBFM ➝ WTW	0.09 *	Accepted	1.875	
*H6*	UFM ➝ WTW	−0.10 **	Accepted	1.785	

Notes: *** *p* ≤ 0.001 (1%), ** *p* ≤ 0.01 (5%), * *p* ≤ 0.05 (10%).

**Table 7 ijerph-18-04577-t007:** Goodness-of-fit indices results.

Term	Value	Recommended Value	Description
CFI	0.973	>0.9 good fit	Comparative fit index
NFI	0.966	>0.9 good fit	Normed fit index
IFI	0.990	>0.9 good fit	Incremental fit index
TLI	0.978	>0.9 good fit	Tucker-Lewis index
GFI	0.994	>0.9 good fit	Goodness of fit index
RMSEA	0.032	<0.08 good fit	Root mean squared error of approximation
X^2^/df	1.381	<3 good fit	Chi-square
SRMR	0.034	<0.09 good fit	Standardized root mean squared residual

**Table 8 ijerph-18-04577-t008:** Endogeneity findings.

Hypotheses	Structural Paths	b Value	*t-*Value	Description
*H1*	ATD ➝ WTW	0.07 **	3.036	Not different
*H2*	SNR ➝ WTW	0.13 **	0.285	Not different
*H3*	CST ➝ WTW	−0.04 ***	−3.445	Not different
*H4*	RPP ➝ WTW	0.08 **	4.272	Not different
*H5*	PBFM ➝ WTW	0.05 *	5.844	Not different
*H6*	UFM ➝ WTW	−0.03 **	−2.758	Not different

Notes: *** *p* ≤ 0.001 (1%), ** *p* ≤ 0.01 (5%), * *p* ≤ 0.05 (10%).

## Data Availability

The data will be made available on reasonable request from the corresponding author.

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
