# Peer review of "Assessing Public Willingness to Wear Face Masks during the COVID-19 Pandemic: Fresh Insights from the Theory of Planned Behavior"

_ijerph, 2021, doi:10.3390/ijerph18094577_

Round 1

Reviewer 1 Report

1. Public behavior is not reflected in the methodological framework and scale of this paper, how to reflect “Assessing Public Behavior” in the Title and Introduction?

2. Line 80, “the behavioral framework of theory of planned behavior (TPB)”, it is necessary to add a reference to indicate the original literature of the TPB.

3. Line 93, descriptions of the original TPB model and its contents should be added.

4. Line 95, “people’s attitudes are shaped by their striking convictions and the outcomes associated with a specific behavior”, the introduction of attitude and the introduction of the other five influencing factors were divided into two paragraphs, the description of these six influencing factors is best placed in one paragraph.

5. Line 99, Figure 1 needs to be revised because it takes up too much space, and Figure 1 is more appropriately placed at the end of section 2.1.

6. Line 102-104, “Researchers have assumed that numerous elements influence the acceptance of a particular product or service in social, economic, and political terms”, the reference should be given to make it more convincible.

7. Line 151, it is more appropriate to replace “Cost” with “Cost of face masks”.

8. Line 167, is “understating” misspelled?

9. Line 173, “Global warming and ever-increasing carbon emissions have become fundamental issues of modern world”, this sentence has nothing to do with the topic.

10. Line 209, in Figure 2, it will be more complete to add the percentage of the sample size for each city to the map.

11. Line 257, why is the first letter of “Valuation” capitalized?

12. Line 323, “Accordingly, hypotheses 3 and 5 were also accepted”, 6 was wrongly written as 5.

13. Line 328, in Figure 4, what do the dashed and solid lines represent respectively? And the arrows in Figure 4 are too dense.

14. Line 403-408, to make the structure of the article clearer, it is better to add a subheading to this paragraph.

15. Line 475, it is not clear what “R&D” means, please provide more explanations.

16. There are still some English grammar mistakes existing in the paper, and a thorough proofread is highly recommended.

Author Response

For improvement on this study, my suggestions are provided as follows:

Comment 1: Public behavior is not reflected in the methodological framework and scale of this paper, how to reflect “Assessing Public Behavior” in the Title and Introduction?

Response to comment 1: Dear reviewer, thank you very much for the positive evaluation of our paper and useful suggestions. We appreciate your time and energy in improving the contents of this study.

We totally agree with your argument; therefore, we have modified the title of the paper by omitting the word “Behavior” to reflect the scale of the paper in accordance with the title. The new title of the paper is “Assessing Public Willingness to Wear Face Masks During the COVID-19 Pandemic: Fresh Insights from the Theory of Planned Behavior”. Changes have been done in the introduction section as well. Please see the modified title of the paper and introduction section. 

Comment 2: Line 80, “the behavioral framework of theory of planned behavior (TPB)”, it is necessary to add a reference to indicate the original literature of the TPB.

Response to comment 2: As demanded, we have provided the reference to indicate the original literature of the TPB as follows:

Ajzen, I. From intentions to actions: A theory of planned behavior. In Action control; Springer, Berlin, Heidelberg: Berlin, Heidelberg, 1985; pp. 11–39.

The above reference has been also provided at the suggested place in the revised manuscript. (Please see line 79).

Comment 3: Line 93, descriptions of the original TPB model and its contents should be added.

Response to comment 3: Thank you very much for the constructive comment to enrich the quality of our paper. Following your suggestion, a detailed description of the original TPB model and its contents have been provided along with a new Figure (Figure 1), displaying the original TPB model. (Please see lines 109–127 and Figure 1).

Comment 4: Line 95, “people’s attitudes are shaped by their striking convictions and the outcomes associated with a specific behavior”, the introduction of attitude and the introduction of the other five influencing factors were divided into two paragraphs, the description of these six influencing factors is best placed in one paragraph.

Response to comment 4: Thank you very much for your valuable comment and suggestion. As suggested, we have merged these two paragraphs for a better presentation of all six influencing factors of public WTW face masks. Please see the revised paper.

Comment 5: Line 99, Figure 1 needs to be revised because it takes up too much space, and Figure 1 is more appropriately placed at the end of section 2.1.

Response to comment 5: We appreciate your valuable suggestion. Following your kind suggestion, we have revised Figure 2 (formerly Figure 1) and shifted it at the end of section 2.1.

Comment 6: Line 102-104, “Researchers have assumed that numerous elements influence the acceptance of a particular product or service in social, economic, and political terms”, the reference should be given to make it more convincible.

Response to comment 6: As demanded, the following references have been provided in this regard to validate the statement.

Wüstenhagen, R.; Wolsink, M.; Bürer, M.J. Social acceptance of renewable energy innovation: An introduction to the concept. Energy Policy 2007, 35, 2683–2691, doi:10.1016/j.enpol.2006.12.001.

Olshavsky, R.W.; Granbois, D.H. “Consumer Decision Making-Fact or Fiction?” Comment. J. Consum. Res. 1980, 7, 331, doi:10.1086/208821

These references have been cited in the revised manuscript as well (Please see line 129).

Comment 7: Line 151, it is more appropriate to replace “Cost” with “Cost of face masks”.

Response to comment 7: We appreciate your kind suggestion. Following your suggestion, we have replaced “Cost” with “Cost of face masks” at the suggested place.

Comment 8: Line 167, is “understating” misspelled?

Response to comment 8: Thank you very much for a thorough review of our paper to improve its quality by indicating the typo mistakes. As suggested, this typo is corrected.

Comment 9: Line 173, “Global warming and ever-increasing carbon emissions have become fundamental issues of modern world”, this sentence has nothing to do with the topic.

Response to comment 9: We are grateful for your suggestion. This sentence has been omitted from the manuscript.

Comment 10: Line 209, in Figure 2, it will be more complete to add the percentage of the sample size for each city to the map.

Response to comment 10: As suggested, the percentage of the sample size for each city is added to the map. Please see Figure 3 (formerly Figure 2).

Comment 11: Line 257, why is the first letter of “Valuation” capitalized?

Response to comment 11: We apologize for the inconvenience. The first capital letter of this word is replaced with small letter in the revised paper.

Comment 12: Line 323, “Accordingly, hypotheses 3 and 5 were also accepted”, 6 was wrongly written as 5.

Response to comment 12: Thank you very much for pinpointing this mistake. We have corrected this mistake in the revised manuscript. Please see line 356.

Comment 13: Line 328, in Figure 4, what do the dashed and solid lines represent respectively? And the arrows in Figure 4 are too dense.

Response to comment 13: We appreciate your query. Dashed lines represent insignificant paths, while solid lines represent significant paths. A better explanation has been also provided in the revised caption of Figure 5 (formerly Figure 4). Besides, the quality of this figure has been further improved for more clarity. Please see the revised Figure 5.

Comment 14: Line 403-408, to make the structure of the article clearer, it is better to add a subheading to this paragraph.

Response to comment 14: As per your kind suggestion, a new subheading “Demographic factors and WTW face masks” is added in this regard. Please see line 452.

Comment 15: Line 475, it is not clear what “R&D” means, please provide more explanations.

Response to comment 15: Thank you very much for your query. The emergence of the novel COVID-19 has compelled people to follow certain laws in order to prevent its transmission among the public. One of these laws is the obligation to wear a facemask at public places. As such, the demand for face masks has escalated recently. Pakistan is a developing country with a high population density. To meet the demand for its large population, the country is facing numerous challenges such as limited manufacturing capacity of face masks, lack of certified suppliers and sellers, low quality of locally manufactured face masks and reliance on foreign countries for the import of face masks. R&D is a vital value-adding segment of the health industry’s value chain. From the perspective of future pandemics, the government of Pakistan should invest resources in R&D to innovate face mask production materials that would incur lower costs, making the availability of low-cost face masks feasible, and satisfy the current needs of ever-increasing population. In this regard, the government should work in collaboration with private institutions and manufacturing companies and devise a solution, which achieves the following three important requirements. (i) The facemasks should be inexpensive, (ii) the facemasks should be sturdy and safe (iii) The facemasks should be washable, sterilizable, and reusable. One classic example is the manufacturing of a three-dimensional reusable facemask, which is being used in several countries. A three-dimensional reusable facemask has the following characteristics. (i) It can be conveniently made on a low-cost, non-heated bed at a low temperature. (ii) It uses a small quantity of filament content. (iii) It can be washed and disinfected, making it reusable. (iv) It uses small quantities of disposable nonwoven cloth, which should be discarded after each use. Another such example is the development of innovative masks (such as degradable masks, reusable masks, and antiviral masks). However, this is impossible to achieve without government support in the form of allocating special funds and subsidies for R&D activities in the long-run. A thorough detail is also provided in the revised paper. (Please see lines 532–554).  

Comment 16: There are still some English grammar mistakes existing in the paper, and a thorough proofread is highly recommended.

Response to comment 16: We appreciate your concern. We have thoroughly proofread the paper by taking the assistance of a professional English language editing service (Springer Nature). As a proof, the editing certificate is attached in the supplementary file. Finally, we have double-checked the entire paper to ensure the English language quality, which the International Journal of Environmental Research and Public Health expects.

Now, the authors are quite confident that the paper in its current revised version is improved to the best possible point compared to the previous version which will satisfy the concerns raised by the respected reviewer. Once again, thank you very much for your professional review of our manuscript. We are looking forward to hearing from you soon.

*************************************************************

Reviewer 2 Report

This paper applies theory of planned behavior (TPB) to study the willingness to wear (WTW) face masks in Pakistan by examining several hypotheses. The factors are preset by the authors and model setting needs more justifications.

  1. It will be good to justify more about the factor choice, and introduce the model in more mathematical or statistical way.
  2. Figure 1 , 2 need be modified to improve the presentation
  3. The total sample size of 738 for several cities not enough to support the findings
  4. The introduction of Theories and presentation of Discussion need to be improved.

Author Response

Comment 1: This paper applies theory of planned behavior (TPB) to study the willingness to wear (WTW) face masks in Pakistan by examining several hypotheses. The factors are pre-set by the authors and model setting needs more justifications.

Response to comment 1: Dear reviewer, thank you very much for the positive evaluation of our paper and useful suggestions. We appreciate your time and energy in improving the contents of this study.

The factors were not pre-set, but they were included after a thorough review of previous literature. In our manuscript, section 2.2 presents the hypothesis development and selection of factors affecting WTW face masks based on an in-depth review of each factor. Moreover, the inclusion of factors affecting consumers’ willingness in the sense of adoption behaviour has been considered by a number of previous studies, which totally justifies the consideration of those factors in the current framework. Thus, the theoretical framework in section 2.1 has been developed based on justifications from the past literature.

Comment 2: It will be good to justify more about the factor choice and introduce the model in more mathematical or statistical way.

Response to comment 2: Thank you very much for your kind comment. Respected reviewer, since the theory of planned behaviour (TPB) has been originally proposed in a theoretical setting and was never presented in the mathematical or statistical model format, we have followed the same tradition. In the revised version, we have further added a TPB diagram with original factors to show how the original factors were linked to the behavioral intentions (see Figure 1). Please take a look at Figure 1 and Figure 2 to have a complete picture of newly added factors. Additionally, we have included your suggestion to the limitation of this work, and future works should develop some mathematical or statistical model in the same domain. (Please see lines 488–491).

Comment 3: Figure 1, 2 need be modified to improve the presentation.

Response to comment 3: We appreciate your time and energy in improving the contents of our paper. As demanded, both these figures have been modified for a better presentation. (Please see the modified figures).

Comment 4: The total sample size of 738 for several cities not enough to support the findings.

Response to comment 4: Thank you very much for your query. The following three criteria were considered while conducting the questionnaire survey. (i) We approached respondents in person and followed a convenience sampling method in the five provincial capital cities of Pakistan. (ii) We conducted the survey during August, September, and October (2020) when the novel coronavirus (COVID-19) was at its peak in Pakistan and it was very difficult to approach so many respondents due to government-imposed restrictions such as lock down, travel constraints, and social distancing policies. Anyhow, we managed to administer 900 questionnaires in the studied areas in such difficult time. However, due to incomplete and missing information, a total of 738 valid responses were collected, with a response rate of 82%. (iii) We followed the Comfrey and Lee’s scale to determine the adequacy of sample size. For instance, Comfrey and Lee (1992) recommended the following scale, (very poor – 50), (poor – 100), (fair – 300), (very good – 500), (excellent – 1000 or more). According to this scale, even the size of our study sample (738 respondents) falls under the "very good" category, ensuring that the sample size is a representative for this research and support its findings.

For further clarification, please see the following reference in this regard.

Comrey, A.L.; Lee, H.B. A First Course in Factor Analysis; 2nd ed.; Lawrence Erlbaum Associates, Inc., Publishers: Hlilsdale, New Jersey 07642, 1992; ISBN 0805810625.

We also agree with the expert reviewer’s opinion that a sample size of 738 is not enough for several cities for the generalizability of the findings. However, it is not possible to expand the sample size at this stage. Therefore, following your argument, we have considered it as a limitation of the study and included in the limitations section of the paper. Subsequent studies can tackle this limitation by expanding the sample size. (Please see lines 484–488).  

Comment 5: The introduction of Theories and presentation of Discussion need to be improved.

Response to comment 5: We are grateful for your constructive suggestion to improve the contents of this study. Following your suggestion, we have improved the introduction of theories and presentation of discussion for the better understanding of readers. The theoretical framework has been improved with the introduction of different theories. In addition, a detailed description of the original TPB model and its contents have been provided along with a new Figure (Figure 1), displaying the original TPB model. (Please see lines 89–116 and Figure 1).

The discussion section is thoroughly improved by supplementing new studies Besides, one new section (COVID-19 post-pandemic agenda) has been added in the discussion section as well to further strengthen the discussion section. (Please see the revised section “5. Discussion”).

After responding to all the concerns raised by the respected reviewer, the authors are quite confident that the current revised paper has been improved to the best possible point compared to the previous version. Once again, thank you very much for your professional review of our manuscript. We are looking forward to hearing from you soon.

*************************************************************

Reviewer 3 Report

The aim of the article submitted by Irfan et al. was to examine public behaviour and willingness to wear facemasks, by analysing the factors that motivate or inhibit people from wearing them. The study is very clear and interesting. Study's details are clearly presented.

However, I have some points that should be checked.

Please, the authors should describe figures 3 and 4 in their caption to better understanding for readers.

What p-values did the authors consider as significant? This information should be added in methods.

It is unclear to me whether the questionnaires were anonymous or not, since the authors say they approached the respondents in person for the survey. Please clarify

In lines 364 and 366, in my opinion the authors should  add in the text the name of the authors of the mentioned references, for a better understanding (as correctly done in line 375)

Author Response

The aim of the article submitted by Irfan et al. was to examine public behaviour and willingness to wear facemasks, by analysing the factors that motivate or inhibit people from wearing them. The study is very clear and interesting. Study's details are clearly presented. However, I have some points that should be checked.

Comment 1: Please, the authors should describe figures 3 and 4 in their caption to better understanding for readers.

Response to comment 1: Respected reviewer, thank you very much for your interest and positive evaluation of our paper. We are grateful for your precious time and energy in reviewing this research article. As suggested, we have provided details of Figures 3 and 4 in their respective captions for the better understanding of readers. Please refer to the revised captions of Figures 4 and 5 (formerly Figure 3 aqnd 4) to see these modifications.

Comment 2: What p-values did the authors consider as significant? This information should be added in methods.

Response to comment 2: Thank you very much for the query. P-values are the probability of obtaining an effect at least as extreme as the one in the sample data, assuming the truth of the null hypothesis. As mentioned in the captions of Figure 5, Table 6, and Table 8, we considered three significance levels, such as 1%, 5%, and 10%. Significance at 1% level (p ≤ 0.001) is indicated by (***), significance at 5% level (p ≤ 0.01) is indicated by (**), while significance at 10% level (p ≤ 0.05) is indicated by (*). Following your suggestion, we have provided this information in the revised paper as well. (Please see lines 347–349).  

Comment 3: It is unclear to me whether the questionnaires were anonymous or not, since the authors say they approached the respondents in person for the survey. Please clarify.

Response to comment 3: Thank you very much for your concern. We followed a convenience sampling method and approached respondents in person for the survey. However, the questionnaires were totally anonymous. Only the demographic details i.e., age, gender, income, education, and occupation were reported in the paper without revealing the anonymity of respondents (such as their names). Please refer to Table 2 in this regard.

Comment 4: In lines 364 and 366, in my opinion the authors should add in the text the name of the authors of the mentioned references, for a better understanding (as correctly done in line 375).

Response to comment 4: We appreciate your valuable suggestion to improve the quality of our paper. Following your suggestion, we have mentioned the names of authors for a better understanding in the revised manuscript.

By responding to expert reviewer’s all valuable comments and incorporating useful suggestions, the authors are quite confident that this paper in the current revised version has been improved to the best possible point compared to the previous version, which will satisfy the concerns raised by the worthy reviewer. Once again, thank you very much for your professional review of our manuscript. We are looking forward to hearing from you soon.

*************************************************************

Round 2

Reviewer 1 Report

The revised paper has met the requirements.

Reviewer 2 Report

I am fine with their revision and has no problem to accept the manuscript now. 

This manuscript is a resubmission of an earlier submission. The following is a list of the peer review reports and author responses from that submission.

Round 1

Reviewer 1 Report

This is a well-designed study based on the preceding studies on people’s attitude in 2009 H1N1 and 2003 SARS-COV-1.

Followings are comments;
Line 27 The survey approach (“in person”, line 214) should be briefly explained.
Line 206 “(see Appendix)”: This should be moved to line 238, as Appendix shows the construction of questions.
Line 214 Preferably, concrete approach to survey (how the respondents were accessed) should be clarified.

Line 215 The term "the convenient random sampling" does not exist. This questionnaire method is the complete convenient sampling. For this reason, in order to compare the population and the sample, the Kolmogorov-Smirnov test between the population and the sample (see https://www.mdpi.com/1660-4601/17/10/3517/htm, Fig.1.) should be conducted to test that there is no difference between the two, and then the next analysis should be conducted.

Line 419-425 “Most importantly … hospitals.”: What findings developed this statement?

Author Response

This is a well-designed study based on the preceding studies on people's attitude in 2009 H1N1 and 2003 SARS-COV-1. Followings are comments;

Comment 1: Line 27 The survey approach ("in person", line 214) should be briefly explained.

Response to comment 1: Thank you very much for the positive evaluation of our paper and useful suggestions. We appreciate your time and energy in improving the contents of this study.

At line 27, the survey approach means "questionnaire survey". The description is also given in the revised manuscript (please see the abstract). While, in-person approach means that we contacted respondents personally and distributed questionnaires to obtain valid and meaningful results during the data gathering stage. Various scholars have used in-person approach for conducting questionnaires in their studies. For instance:

Reuter, K.E.; Schaefer, M.S. Illegal captive lemurs in Madagascar: Comparing the use of online and in-person data collection methods. Am. J. Primatol. 2017, 79, 22541, doi:10.1002/ajp.22541.

Shapka, J.D.; Domene, J.F.; Khan, S.; Yang, L.M. Online versus in-person interviews with adolescents: An exploration of data equivalence. Comput. Human Behav. 2016, 58, 361–367, doi:10.1016/j.chb.2016.01.016.

Woodyatt, C.R.; Finneran, C.A.; Stephenson, R. In-Person versus online focus group discussions: A comparative analysis of data quality. Qual. Health Res. 2016, 26, 741–749, doi:10.1177/1049732316631510.

The above references have also been incorporated in the revised manuscript to support our arguments.

Comment 2: Line 206 "(see Appendix)": This should be moved to line 238, as Appendix shows the construction of questions.

Response to comment 2: As demanded, we have moved "(see Appendix)" to the suggested place. (Please see page 7, line 257).

Comment 3: Line 214 Preferably, concrete approach to survey (how the respondents were accessed) should be clarified.

Response to comment 3: Thank you very much for the constructive comment to enrich the quality of our paper. Before the conduction of survey, the authors visited the provincial capitals of Pakistan (Lahore, Peshawar, Karachi, Gilgit, and Quetta) to find the distinguishing features of participants, living in these cities to find the distinctive characteristics of participants, living in these cities. After that, the respondents were approached in-person (contacted personally) for the actual questionnaire survey. The following criteria were considered for the selection of respondents.

(i) The respondents should be the permanent residents of these cities.

(ii) The age of respondents should be not less than 18 years. 

The questionnaire conduction process was divided into two phases. During the first phase, questionnaires were handed over to 900 respondents, and they were given a time period of one month to fill their responses. An in-depth explanation was provided to participants about every element of the questionnaire to obtain accurate and meaningful results. During the second phase, questionnaires were taken back from respondents after one month. A total of 738 valid responses (the questionnaires whose all questions were filled by the respondents) were collected with a response rate of 82%. (Please see page 6, lines 215–237).

Comment 4: Line 215 The term "the convenient random sampling" does not exist. This questionnaire method is the complete convenient sampling. For this reason, in order to compare the population and the sample, the Kolmogorov-Smirnov test between the population and the sample (see https://www.mdpi.com/1660-4601/17/10/3517/htm, Fig.1.) should be conducted to test that there is no difference between the two, and then the next analysis should be conducted.

Response to comment 4: Thank you very much for your valuable comment and suggestion. Firstly, we agree with your opinion that the sampling process was not purely randomized. Rather, a convenient sampling approach was followed due to the ongoing epidemic outbreak. Previous studies have also adopted this sampling technique. For instance:

https://doi.org/10.1016/j.egyr.2017.03.002

https://doi.org/10.1007/s11596-015-1407-4

https://doi.org/10.11648/j.ajtas.20160501.11

Generally, this sampling method, due to convenience and feasibility, is useful for researchers in some special situations like epidemics or experimental behavioral research. Therefore, it challenges the perfect generalizability of the empirical findings based on the drawn sample. However, the respondents' demographic features have shown that the questionnaires were conducted from the respondents belonging to heterogeneous backgrounds. Thus, the findings generated based on such a sample provided a fair representation of the population with heterogeneous backgrounds in terms of education, age, income, and occupation. Moreover, the questionnaire survey was conducted in all Pakistan's capital cities, involving respondents from a population with diverse cultures and behaviors. Along these lines, the generated sample was rich enough to draw a satisfactory representation of a population with heterogeneous features. (Please see page 6, lines 220­–231).

Secondly, the expert reviewer suggested the application of the Kolmogorov-Smirnov test between the population and the sample. This test was employed by Hashiguchi et al. (2020). For instance:

https://doi.org/10.3390/ijerph17103517

They categorized the construction workers based on the only demographic feature of "age". They compared the percentage of construction workers in the actual population to the sampled workers' percentage for different age groups. However, in the present case, the application of the Kolmogorov-Smirnov test is not feasible. The reason is that the population data for the heterogeneous demographic features are unavailable to compare with the sampled respondents' features. This situation limits the perfect generalizability of empirical findings for the whole population. Therefore, we have included this point in the limitations of the current research, which was the consequence of not being able to conduct random sampling during the severe outbreak of pandemic across the country. Further, we have modified the mentioned statement from "convenient random sampling" to "convenient sampling." Please refer to the revised manuscript to see these modifications. (Please see page 14, lines 437­–445).

The references mentioned above have been incorporated in the revised manuscript along with additional relevant references to further strengthen our study. For instance:

Hashiguchi, N.; Cao, J.; Lim, Y.; Kubota, Y.; Kitahara, S.; Ishida, S.; Kodama, K. The effects of psychological factors on perceptions of productivity in construction sites in Japan by worker age. Int. J. Environ. Res. Public Health 2020, 17, 3517, doi:10.3390/ijerph17103517.

Hashiguchi, N.; Sengoku, S.; Kubota, Y.; Kitahara, S.; Lim, Y.; Kodama, K. Age-Dependent influence of intrinsic and extrinsic motivations on construction worker performance. Int. J. Environ. Res. Public Health 2020, 18, 111, doi:10.3390/ijerph18010111.

Zhou, D.; Shah, T.; Jebran, K.; Ali, S.; Ali, A.; Ali, A. Acceptance and willingness to pay for solar home system : Survey evidence from northern area of Pakistan. Energy Reports 2017, 3, 54–60, doi:10.1016/j.egyr.2017.03.002.

Etikan, I.; Musa, S.A.; Alkassim, R.S. Comparison of convenience sampling and purposive sampling. Am. J. Theor. Appl. Stat. 2016, 5, 1, doi:10.11648/j.ajtas.20160501.11.

Tang, S.F.; Wang, X.; Zhang, Y.; Hou, J.; Ji, L.; Wang, M.L.; Huang, R. Analysis of high alert medication knowledge of medical staff in Tianjin: A convenient sampling survey in China. J. Huazhong Univ. Sci. Technol. - Med. Sci. 2015, 35, 176–182, doi:10.1007/s11596-015-1407-4.

Comment 5: Line 419-425 "Most importantly … hospitals.": What findings developed this statement?

Response to comment 5: Thank you very much for your valuable query. The stated arguments were not clear enough to extend robust policy outcomes of this work. Therefore, we have further modified the policy proposal stemming from the findings of this research. These policies are proposed as follows:

Research results revealed that face masks' cost and unavailability are critical barriers to the public's willingness to wear them; therefore, robust policy development is needed to overcome these impediments. To this end, on the one hand, the federal and provincial governments should subsidize the import of face masks. On the other hand, the availability of face masks should be ensured by channeling the local medical stores' systematic monitoring. An alternative option to reduce the consumer price of face masks could be the subsidization of local manufacturers. Rapid domestic production of the face masks will curtail their cost for the end-users. Moreover, enhanced provision of the face masks is expected to increase public willingness to wear them. Besides, from the perspective of the future pandemics, the governments should invest resources into research and development (R&D) to innovate face masks production materials that would incur a lesser cost, making the availability of inexpensive face masks feasible. Additionally, risk perception strongly drove the public willingness to wear face masks; therefore, the campaigns on lethality and fatality of the pandemic should be conducted at all societal levels to better perceive the pandemic's risk. In this way, enhanced credibility of the pandemic threat will promote the willingness to wear face masks.

We have also included the above-modified arguments in the revised manuscript. Please refer to the revised manuscript to see these modifications. (Please see pages 14,15, lines 479–492).

Now, the authors are quite confident that the introduction section in the current revised version has been improved to the best possible point compared to the previous version of the manuscript. Once again, thank you very much for your professional review of our manuscript. We are looking forward to hearing from you soon.

*************************************************************

Reviewer 2 Report

Dear authors, I read carefully your paper. It is interesting and scientifically well-sounding. However, some points should be clarified and amended:

  1. The introduction should be more concise. Some paragraphs about literature (lines 71- 80) data should be transferred in the discussion, so that the objective of the work is more highlighted;
  2. Please clarify the recruitment criterion of respondents (where and in which way did you contact participants? In which setting?);
  3. You considered valid 82% of responses. Please clarify the criteria to consider the response as valid;
  4. In results, you did many efforts to check the internal and external validity of your results, and this is correct. You stated that your hypotheses are satisfied. In addition to this “internal” analysis, it would be very useful, from a practical point of view, to understand which of the variables examined has the most decisive impact (positively or negatively) with the willingness to wear (WTW) face masks. You analyzed in your results the strength of positive and negative associations between each variable with WTW, but this data, that is essential for an external reader, is not sufficiently highlighted in the discussion and conclusion. This information will improve the practical value of your paper, indicating what are the factors to be given greater importance in public health policies and communication.

Author Response

Dear authors, I read carefully your paper. It is interesting and scientifically well-sounding. However, some points should be clarified and amended:

Comment 1: The introduction should be more concise. Some paragraphs about literature (lines 71- 80) data should be transferred in the discussion, so that the objective of the work is more highlighted;

Response to comment 1: Thank you very much for the positive evaluation and feedback of our paper. According to your kind suggestion, we have concise the introduction section and the said paragraph has been moved to the discussion section to clearly highlight the objective of the study. (Please see page 13, lines 385­­–388 and lines 415–419).

Comment 2: Please clarify the recruitment criterion of respondents (where and in which way did you contact participants? In which setting?);

Response to comment 2: We appreciate your time and energy in improving the contents of our paper. Before the conduction of the survey, the authors visited the provincial capitals of Pakistan (Lahore, Peshawar, Karachi, Gilgit, and Quetta) to find the distinguishing features of participants, living in these cities. After that, the respondents were approached in-person (contacted personally) for the actual questionnaire survey. The following criteria were considered for the selection of respondents.

(i) The respondents should be the permanent residents of these cities.

(ii) The age of respondents should be not less than 18 years.

The sample was generated using the convenient sampling method. The sample indicates a balanced ratio from every life sector, as we selected respondents from all groups of income, age, education, and occupation. The questionnaire conduction process was divided into two phases. During the first phase, questionnaires were handed over to 900 respondents, and they were given a time period of one month to fill their responses. An in-depth explanation was provided to participants about every element of the questionnaire to obtain accurate and meaningful results. During the second phase, questionnaires were taken back from respondents after one month and valid responses were included in the statistical analysis (Please see page 6, lines 215–236).

Comment 3: You considered valid 82% of responses. Please clarify the criteria to consider the response as valid;

Response to comment 3: Thank you very much for the valuable suggestion to enrich the contents of this study. The following three criteria were followed to consider the response as valid.

(i) The questionnaires whose all aspects were thoroughly filled by the respondents.

(ii) The questionnaires which do not have any missing or incomplete information.

(iii) Finally, the questionnaires which do not have multiple responses.

The explanation has also been provided in the revised manuscript. (Please see page 6-7, lines 237–240).  

For further clarification, please see the newly added references in this regard.

Hägerhed Engman, L.; Bornehag, C.G.; Sundell, J. How valid are parents' questionnaire responses regarding building characteristics, mouldy odour, and signs of moisture problems in Swedish homes? Scand. J. Public Health 2007, 35, 125–132, doi:10.1080/14034940600975658.

Tauni, M.Z.; Fang, H.X.; Rao, Z. u. R.; Yousaf, S. The influence of investor personality traits on information acquisition and trading behavior: Evidence from Chinese futures exchange. Pers. Individ. Dif. 2015, 87, 248–255, doi:10.1016/j.paid.2015.08.026.

The above references have been included in the revised paper to support our arguments.

Comment 4: In results, you did many efforts to check the internal and external validity of your results, and this is correct. You stated that your hypotheses are satisfied. In addition to this "internal" analysis, it would be very useful, from a practical point of view, to understand which of the variables examined has the most decisive impact (positively or negatively) with the willingness to wear (WTW) face masks. You analyzed in your results the strength of positive and negative associations between each variable with WTW, but this data, that is essential for an external reader, is not sufficiently highlighted in the discussion and conclusion. This information will improve the practical value of your paper, indicating what are the factors to be given greater importance in public health policies and communication.

Response to comment 4: Thank you very much for your constructive comment and suggestion. Following your suggestion, we have further improved the discussions to include the importance of the variables based on the empirical results. For this purpose, a new sub-section 5.7 entitled "Summary and limitations of research" has been added. To illustrate, we categorize the contributors (variables) into positive and negative. Among the positive contributors to WTW face masks, risk perception demonstrated the most substantial contribution. Its intuition is that the better the risk is perceived; more people will be ready to use face masks. Thus, a lack of risk perception behaviour might lead to contradictory actions. Therefore, steady efforts to make the people aware of the pandemic's fatality and lethality would keep on improving the risk perception, positively impacting the WTW face masks. On the contrary, public attitude and perceived benefits of face masks remained the least contributors to promoting the WTW face masks. It implies that people respond to the benefits of face masks with less intensity as compared to threats of not wearing them amid the pandemic outbreak such as COVID-19. Among the negative contributors, the cost of face masks delivered an almost negligible but significant contribution. It indicated the importance of cost in shaping the consumers' purchase decisions of the face masks. Contrariwise, face masks' unavailability proved to be a negative and relatively stronger contributor to the public willingness to wear them. It depicted the actual scenario of the country. Amid the pandemic outbreak, a shortage of face masks' supply was observed across the country. Hence, the unavailability of face masks put forward a practical obstacle. To overcome this hurdle, the supply of face masks should be enhanced to increase the willingness to purchase and wear them.

We have included the above arguments in the newly added sub-section 5.7 and conclusions part of the revised manuscript. Please refer to the revised manuscript to see these modifications. (Please see pages 13-14, lines 420­–436).

Now, the authors are quite confident that all sections in the current revised version have been improved to the best possible point compared to the previous version of the manuscript. Once again, thank you very much for your professional review of our manuscript. We are looking forward to hearing from you soon.

*************************************************************

Round 2

Reviewer 1 Report

Thank you very much for your very polite revise. I think you have responded politely to my comments. I agree with the publication of this article.